# Impact of a 5-Week Individualised Training Program on Physical Performance and Measures Associated with Musculoskeletal Injury Risk in Army Personnel: A Pilot Study

**DOI:** 10.3390/sports11010008

**Published:** 2023-01-03

**Authors:** Chelsea Smith, Kenji Doma, Brian Heilbronn, Anthony Leicht

**Affiliations:** 1College of Medicine and Dentistry, James Cook University, Townsville, QLD 4811, Australia; 2Australian Army, Townsville, QLD 4814, Australia; 3Sport and Exercise Science, James Cook University, Townsville, QLD 4811, Australia; 4Australian Institute of Tropical Health and Medicine, James Cook University, Townsville, QLD 4811, Australia

**Keywords:** military, exercise, resistance training, muscular strength, countermovement jump

## Abstract

Objective: To examine the feasibility and effect of an individualised and force-plate guided training program on physical performance and musculoskeletal injury risk factors in army personnel. Design: Pre-post, randomised control. Methods: Fourteen male and five female Australian Army soldiers were randomised into two groups and performed 5-weeks of physical training. The control group (*n* = 9) completed standard, group-designed, physical training whilst the experimental group (*n* = 8) completed an individualised training program. Physical (push-ups, multi-stage fitness test, three repetition maximum (3RM) for squat, strict press, deadlift and floor press), occupational (weight-loaded march time), and technological assessments (two-leg and one-leg countermovement jumps (CMJ), one-leg balance, one-arm plank) were conducted prior to and following the training period. Comparisons between groups and changes within groups were conducted via Mann–Whitney U tests. Results: Compared to the control group, the experimental group exhibited a significantly smaller improvement for weight-loaded march time (−0.7% ± 4.0% vs. −5.1% ± 3.0%, *p* = 0.03) and a greater improvement for deadlift-3RM (20.6% ± 11.9% vs. 8.4% ± 6.8%, *p* = 0.056). All other outcomes were similar between groups. Visually favourable alterations in the two-leg CMJ profile with no reports of injuries were noted for the experimental group. Conclusions: Individualised physical training was feasible within an army setting and, for the most part, produced similar physical, occupational and technological performances to that of standard, group-designed physical training. These preliminary results provide a foundation for future research to expand upon and clarify the benefits of individualised training programs on long-term physical performance and injury risk/incidence in active combat army personnel.

## 1. Introduction

Physical activity is pivotal to the general health of adults [1], as well as to the improvement of readiness, performance and long-term health of army personnel [2,3]. Further, physical activity plays an important role in the prevention of musculoskeletal injury (MSKI) for this vulnerable military population [2,3]. In the Australian Defence Force (ADF), 40% of clinical presentations are attributed to preventable MSKI [4]. Factors that place army personnel at risk of MSKI include poor muscular strength, balance, muscle asymmetry and aerobic fitness [5,6,7]. Given the high volume of running, load carriage and other strenuous activities performed by soldiers, there is great potential for MSKI with the aforementioned risk factors strongly associated with increased MSKI risk [5,7,8].

Traditionally, the physical performance of military personnel has been assessed via running and generic activities (e.g., sit ups, push ups, etc.). More recently, military organisations have begun to adopt more occupationally relevant tests of physical performance such as load carriage and manual handling [9,10,11]. Whilst this is a positive step in assessing the ability of personnel to perform essential occupational tasks, these assessments often provide limited insight into individual movement characteristics (e.g., symmetry) and opportunities to address injury risk factors [12]. Further, military personnel undertake a wide array of physical training including generic aerobic and strength activities, bodyweight training, functional fitness training, and strength training with free weights to enhance performance and minimise injury [13,14]. Whilst studies have confirmed that generic training improves the initial fitness of army personnel and may reduce MSKI risk [8,12], these practices do not target other MSKI risk factors such as balance and strength asymmetry [14]. Therefore, generic training may fail to adequately capitalise on opportunities to mitigate MSKI risk and preserve occupational performance. Alternate training regimes may provide greater attention to MSKI risk factors and lead to greater occupational performance with reduced MSKI.

Pre-screening assessments, which aim to identify MSKI risk factors, are essential to inform the development and implementation of appropriate injury prevention programs [5,7,14]. The use of force plates [15,16,17] and computerised systems in the prevention of MSKI have been of interest recently in both sporting and military settings [1,15,18] with these technologies reported to be more reliable than traditional testing methods (e.g., Vertec jump test) in clarifying performance and identifying injury risk within a military setting [19,20]. One such system utilises a countermovement jump (CMJ) performance via force plates and proprietary software to identify imbalances that may exist (i.e., Sparta Science) [21,22,23]. Based on the CMJ outcomes, which have demonstrated moderate-to-excellent test–retest reliability in military personnel (ICC 0.67–0.91) [17], and implied imbalances, an individualised strength program is then recommended to improve these imbalances and decrease the risk of future MSKI [18,22,23]. To our knowledge, the effectiveness of an individualised and force plate guided program has not been examined in terms of feasibility, physical performance and changes in MSKI risk. The aim of this study was to examine the feasibility and effects of force plate technology to support an individualised training program in reducing MSKI risk (as defined by muscular strength, balance, muscle asymmetry and cardiorespiratory fitness). Clarification of the benefits of a novel, individualised, force plate guided program on performance and MSKI risk would significantly aid army personnel in terms of health, occupational productivity and healthcare costs.

## 2. Methods

### 2.1. Study Design

This study employed a pre-post, randomised control design of a convenient sample of healthy, Australian Regular Army, combat personnel. Participants were assessed for CMJ, balance, core stability, inter-limb asymmetry, muscular strength and endurance, cardiorespiratory fitness and occupational capacity prior to (Week 0) and following (Week 6) a 5-week training intervention. Following the pre-training assessments, participants were randomly block-assigned (via combat unit) into one of two groups using a computer-generated technique (https://www.randomizer.org/ (accessed on 3 March 2020)). The control group (CON) undertook standard military physical training whilst the experimental group (EXP) undertook pre-defined, individualised physical training based on measures obtained from a CMJ test.

### 2.2. Participants

Seventeen healthy, active personnel (13 male, 4 female) volunteered for and completed this study from a range of infantry, artillery, cavalry, communications and military police units. Nine (7 male, 2 female) of these participants were allocated to CON and eight participants (6 male, 2 female) were allocated to EXP. Participants’ initial mean (±standard deviation, SD) age, height, mass and body mass index (BMI) were similar between groups and presented in Table 1. All participants were fully qualified within their individual employment categories and of a deployable medical employment classification. Participants completed a pre-screening form to confirm their healthy status (e.g., free from musculoskeletal injury). Leg and arm dominance were identified as the limb that participants used for kicking and writing/throwing, respectively. Prior to testing, all participants provided written informed consent in accordance with organisational ethical approval (Departments of Defence & Veterans Affairs Human Research Ethics Committee 290-020).

### 2.3. Procedures

Participants completed assessments prior to and following the training program at approximately the same time of the day (06:00–09:00) within the same week (i.e., over 5 days). Participants completed trials of all assessments before the commencement of the study and each assessment was demonstrated prior to each testing session.

### 2.4. Warm Up

On day one, all participants completed a standardised warm-up followed by assessments of lower limb power, core strength/stability, balance, upper body muscular endurance and cardiorespiratory fitness within 1-h. The standardised warm-up lasted 5 min and consisted of the following activities in order: five lateral lunges per leg, five reverse lunges per leg, ten good mornings, ten pogo jumps, five squat jumps, five tuck jumps, ten pogo jumps, five squat jumps, and five tuck jumps.

### 2.5. Countermovement Jump Assessments

After a 2 min rest, participants completed a two-leg CMJ, one-arm plank, one-leg balance and one-leg CMJ assessment without shoes (i.e., barefoot) using a commercially available and calibrated piezoelectric force plate, sampling at a frequency of 1000 Hz (Bertec/Sparta, SSFP01; Sparta Science, Menlo Park, CA, USA) [17]. All participants were instructed to perform each assessment to the best of their ability (i.e., maximal jump height, minimal movement during balance/plank). For the two-leg CMJ, each participant stood on the force plate. Upon hearing an auditory cue that indicated stabilisation of body weight, participants performed a CMJ with 2-arm swing. Participants completed a total of four, maximal-effort two-leg CMJ with 15 s rest intervals provided between successive jumps [17]. A similar procedure was completed by participants during the last assessment of the session for one-leg CMJ assessments of both dominant and non-dominant legs.

### 2.6. Balance Assessments

Participants performed four, one-leg balance tests (two per limb) on the force plate. To establish a baseline reading, each participant was instructed to stand on the force plate with two feet and eyes closed. The first auditory cue indicated to the participant to balance on their right leg (lifting the leg) for 20 s. A second auditory cue signalled the end of the 20 s trial and the participant stepped off the force plate to reset. The procedure was repeated on the left leg. During each balance test, participants were permitted to touch the ground with their elevated foot, if needed briefly, to maintain balance in accordance with the manufacturer’s protocols [17].

### 2.7. Plank Assessments

Participants performed a one-arm plank test with the force plate. Participants stood on the force plate with both feet to allow assessment of body mass. The participant was then prompted to reposition into a push-up/plank position with both hands and feet shoulder-width apart. An auditory cue indicated to the participant to balance on their right hand (picking up the left arm) for 20 s. A second auditory cue signalled the end of the 20 s trial with the participant removing both hands from the force plate and resetting. The procedure was repeated for the left arm, and then repeated for both arms for a total of four trials (two per limb) [17].

### 2.8. Push-Ups and Multistage Shuttle Test

After 5 min of rest, participants’ upper body muscular endurance was assessed as the maximum number of push-ups completed within 1 min [7]. Lastly, participants completed the multistage shuttle test where they were instructed to complete as many 20 m shuttles as possible in accordance with an audio recording [24]. The total number of shuttles completed was recorded with the corresponding measure of cardiorespiratory fitness determined [24].

### 2.9. Muscular Strength Assessments

Approximately 24- and 72 h later, participants were assessed for muscular strength of the legs (via squat) and chest (via bench press), and legs/back (via deadlift) and shoulders (via strict press), respectively. The strength assessments employed a standardised three repetition maximum (3RM) protocol [10]. Participants completed ten repetitions of the test with 50% of their estimated 3RM followed by an additional warm-up set of three-to-five repetitions. Weight was then added in increments of 5–10% until participants attained a 3RM for each movement [10]. A minimum of 3 min of rest was employed between successive attempts.

### 2.10. Weight-Loaded March

Finally, on day five, occupational performance was assessed during a weight-loaded march (i.e., load carriage ability). Following a low-intensity warm-up, participants completed a 5 km march in uniform and boots, carrying a combined weight of 24 kg including training rifle (4 kg) and field backpack (20 kg). Participants completed the weight-loaded march as fast as possible by walking and shuffling [25].

### 2.11. Outcome Measures

The primary outcome measures included the following physical and occupational assessments: number of push-ups in 60-s; predicted rate of oxygen consumption (millilitre/kilogram/minute); squat-3RM (kilogram); strict press-3RM (kilogram); deadlift-3RM (kilogram); floor press-3RM (kilogram) and; 5 km weight-loaded march time (minutes). The secondary outcomes measures included the greatest Sparta score (a population normalised T-score) for the force plate assessments. The two-leg and one-leg CMJ measures included the total Sparta score, jump height (derived from flight time), injury risk score, load score, explode score and drive score. The latter three scores (i.e., load, explode and drive) were derived from the average eccentric rate of contraction (AERC), average concentric force (ACF) and concentric vertical impulse (CVI) during the CMJ [22]. The load, explode and drive scores were also represented visually to produce a “Sparta profile” which allowed for easy identification of participants’ imbalances by practitioners to then inform training [22]. The one-leg balance and plank measures included the Sparta scores for dominant and non-dominant legs and arms, respectively, and the asymmetry ratio between dominant and non-dominant limbs [26,27]. All secondary outcomes were calculated via custom-designed software (Sparta Home 2.0, Sparta Science, Menlo Park, CA, USA) with the population normalised T-scores incorporated into a predictive algorithm to calculate the participant’s injury risk score [22,23,28]. Following the 5-week study period, participants were asked to complete an online survey to document any injuries experienced and their management during the study period. For this study, an injury was defined as “any damage to the body that resulted in reduced occupational or training function.”

### 2.12. Physical Training

Training consisted of one cardiorespiratory session on Monday, strength sessions on Tuesday and Thursday (separated by at least 48 h), and one occupational session on Friday each week for 5-weeks. All groups were prescribed the same interval-based cardiorespiratory running program [i.e., Week 1 = 4 × (400 m at target pace with 1 min rest, 200 m at target pace with 0.5 min rest and 100 m recovery jog); Week 2 = 2 × (4 × 400 m at target pace with 0.5–1 min rest) and 3 min between sets; Week 3 = 4 × (600 m at target pace with 0.5–1 min rest, 200 m at target pace with 1 min rest); Week 4 = 2 × (3 × 600 m at target pace with 0.5–1 min rest) and 3 min between sets; Week 5 = 4 × 800 m at target pace with 0.5–1 min rest; target pace was based upon their individual target time for a 2.4 km basic fitness test)]. All groups were prescribed the same occupational training program consisting of a 24 kg weight-loaded march starting at 5.5 km in Week 1 that increased progressively to 7 km by Week 5. The strength training sessions differed between groups with the CON group undertaking resistance and circuit training. Resistance training included four sets of 8–12 repetitions for back squat/strict press and deadlift/bench press, plus four continuous rounds of 10–12 repetitions of assistances exercises (e.g., sit-ups, upright rows, etc.) at a rating of perceived exertion [RPE] of 7–8 out of 10. Circuit training included 6–20 × 15–30 s sprints with 15–30 s rest periods.

For the EXP group, the strength sessions consisted of activities recommended by the Sparta Science software, based upon participant’s individual CMJ results and unique Sparta profile (e.g., low load, low explode, etc.). Specifically, participants completed six exercises (3 × 2 exercises superset) per day with each exercise consisting of two warm-up and three working sets. Each working set consisted of 3–20 repetitions (pending exercise) at a RPE of 6–10, as recommended for each individual. For example, participants with a low load Sparta profile undertook five sets of front squats (RPE of 4–9, 10–3 repetitions) superset with depth drop (5–3 repetitions of 20–40 cm), five sets of 1-leg squats (RPE of 4–8, 10–8 repetitions) superset with 1-leg calf raises (RPE of 2–6, 20–15 repetitions), and standing barbell overhead press (RPE of 4–9, 10–3 repetitions) superset with bent over row (RPE of 4–8, 15–8 repetitions), all on a 3 min cycle within the same session.

All sessions were progressively altered to enhance training stimulus over the study period. Details of all training sessions completed were documented by participants in a training diary that included session type, duration, load/intensity [29] with a minimum compliance of 50% confirmed for all participants.

### 2.13. Statistical Analysis

Data is presented as mean ± standard deviation (SD), unless otherwise stated. Due to the sample size of each group, comparisons between groups prior to training (Week 0), and changes over time within groups (i.e., percentage change between Weeks 0 and 6) were conducted via Mann–Whitney U tests (IBM SPSS Statistics for Windows, Version 28.0. Armonk, NY, USA). Magnitude of change was calculated as an effect size (ES) using the Mann–Whitney Z statistic and interpreted as large (0.5), medium (0.3), and small (0.1) [30]. The level of significance for all analyses was set at *p* ≤ 0.05.

## 3. Results

Training compliance was similar between groups (65.6% ± 13.6% vs. 71.8% ± 13.2%, *p* = 0.135) with the overall completion being 68.8% ± 13.4% of total possible sessions.

Table 2 summarises the physical and occupational assessments prior to and following the 5-week training period. Both groups exhibited similar physical and occupational performances prior to the training program except for the 5 km weight-loaded march time, which was significantly faster for the EXP group (*p* = 0.038, large ES, Table 2). No significant training changes for physical and occupational performances were observed except for 5 km weight-loaded march time, which was significantly increased for CON compared to EXP (~5% vs. ~1%, *p* = 0.030, large ES, Table 2). The EXP group improved their deadlift-3RM with this change greater compared to CON (~21% vs. ~8%, *p* = 0.054, large ES, Table 2).

Results for the Sparta Science assessments are shown in Table 3 with no significant differences between groups at Week 0 or training-induced changes (small-medium ES). In terms of the two-leg CMJ Sparta profiles, visually apparent differences were observed after the 5-week training period. For CON, the two-leg CMJ Sparta profile changed from a relatively even profile (i.e., similar load, explode and drive scores) at Week 0 to a predominantly low-drive profile at Week 6 (Figure 1). Conversely, the EXP displayed a high-drive profile at Week 0 that varied to a relatively even profile at Week 6 (Figure 1). Based upon the proprietary profile definitions, a total of two participants in CON (22.2%) and five participants in EXP (62.5%) changed their profile from Week 0 to Week 6.

A total of three injuries (all in CON) were reported during the study period. These included aggravations of a prior shoulder injury during training, lower back injury during the weight-loaded march and lower back injury during training. All injuries were managed conservatively (i.e., no medical referral) with rest (i.e., one to seven days of limited activity).

## 4. Discussion

The aim of this pilot study was to determine the feasibility of implementing an individualised and force plate guided physical training program in an active combat army unit. Overall, the individualised program (EXP) was successfully completed and, for the most part, resulted in similar physical, occupational and Sparta Science outcomes to that of standard military physical training (CON). Further, the results indicated potential for force plate technology to support more individualised training prescription in a combat brigade environment. These preliminary results provide a foundation for future research to expand upon and clarify the benefits of: (1) an individualised physical training program on long-term physical performance and injury risk/incidence; and (2) force plate and computerised systems as a monitoring tool to aid exercise prescription for active-duty army personnel.

The arduous nature of the military service is well-known with soldiers expected to complete a range of occupational tasks [14,25]. Subsequently, physical training for the military must be extensive and robust to prepare the war fighter for both acute and chronic occupational physical demands. Recently, a systematic review identified that non-traditional military physical training (i.e., advanced or structured resistance training programs) had a greater effect on performance across a range of fitness domains including occupationally specific tasks compared to traditional military physical training [14]. Furthermore, two recent studies in Australian Army recruits and soldiers have challenged the traditional bias in military physical training towards field expediency and shown it is possible to incorporate progressive resistance training into group training delivery models [25,31]. Such departures away from traditional, ‘one size fits all’ training regimes have also been noted in elite sport with individualised injury prevention training programs recommended [22,23,25]. Whilst such training regimes may be novel for the military, it remains vital that these programs, at the very least, maintain the physical standards required to prepare soldiers for operational demands [14,25]. The current results confirmed that individually designed training resulted in similar physical performance standards as traditional group-based training and therefore was not a significant risk to physical preparation. Further, we noted greater improvements in a measure of muscular strength (deadlift-3RM) for the individualised training group (EXP) compared to group-based training (CON). Most participants in EXP exhibited a low-explode (i.e., ACF) profile for the two leg CMJ (Table 3, Figure 1) and poor trunk/torso stability [18]. Consequently, their training regimes were prescribed with core/trunk exercises including deadlifts to address the limitations identified by the proprietary software [18]. The significant improvement in deadlift-3RM for EXP provides preliminary support for the utility of individualised training in the military to address imbalances and potentially address fitness domains associated with lower MSKI risk [18]. In addition, visually apparent (non-statistical) changes were observed for the two-leg CMJ jump profiles for each group. Force plate and computerised systems include visual profiles to guide practitioners’ identification of individual’s weaknesses that require address through physical training [15]. In the current study, EXP exhibited a high-drive (CVI) and low-explode (ACF) profile at Week 0 that became more even (i.e., similar load, explode and drive scores) by Week 6 (Figure 1). This non-statistical profile change provided preliminary support of the benefits for individualised training programs to tackle disparities identified by force plate technology [18]. In contrast, CON exhibited an initial two-leg CMJ even profile that was altered to a predominantly low-drive (CVI) profile (Figure 1), a profile that has been associated with an increased risk of quadricep and hamstring strain in high school, college and professional athletes [18,22]. Such low power during the CVI phase of the CMJ and potentially greater MSKI risk may be further exacerbated in the military during stressors such as marching over uneven terrain and heavy load carriage that may require consideration during training prescription [14,25,32]. Despite the current pilot results, the association between CMJ metrics and the likelihood of sustaining a MSKI remains unclear [16] with larger, long-term studies needed to clearly extend upon the current study results.

The current study has made a novel and preliminary contribution to our understanding of the potential for force plate technology to support more individualised training for military personnel. To our knowledge, the current study has been the only one that has examined the impact of an individualised training program based on MSKI risk factors on physical performance measures in army personnel. Prior studies of individualised training programs have focused on performance optimisation for elite athletic populations, rather than MSKI risks, with extremely different physical training requirements to the military [18]. Secondly, the current study incorporated active-duty soldiers within a military setting to genuinely assess the real-world feasibility and application of individualised training programs [3]. Thirdly, fitness measures, military-specific assessments and technology-derived metrics (i.e., Sparta scores) were assessed concurrently to comprehensively assess the impact of tailored training programs from a physical performance, occupational and technological perspective [7,22,32].

Notwithstanding these unique strengths, several limitations of the current study existed. Firstly, a modest military cohort (*n* = 17) was engaged in this pilot study due to occupational demands/duties. Secondly, the duration of the training period was shorter than originally planned (5-weeks instead of 10-weeks) due to the occupational demands/duties of an active combat brigade. Future research incorporating larger sample sizes and longer training durations are encouraged to clarify the long-term benefits of individualised training physical performance and injury risk for army personnel [33,34]. Thirdly, the degree of heterogeneity of the participants in this study was high in terms of fitness levels and occupational demands. This being said, the participants examined in the current study were reflective of the diverse nature of a combat brigade, mimicking the potential for an emerging technology to be applied to a real-world military setting.

## 5. Conclusions

In conclusion, the current pilot study identified that a program of individualised physical training was feasible and produced similar physical, occupational and Sparta Science performances to that of standard, group-designed, physical training in military personnel. The results of the current study indicate that force plate technology may have utility in supporting more individualised physical training programs to improve occupational performance and/or reduce injury risk profiles. Future research is recommended to elucidate the health, occupational productivity and healthcare cost benefits of individualised training on performance and MSKI risk in larger samples of military personnel.

## Figures and Tables

**Figure 1 sports-11-00008-f001:**
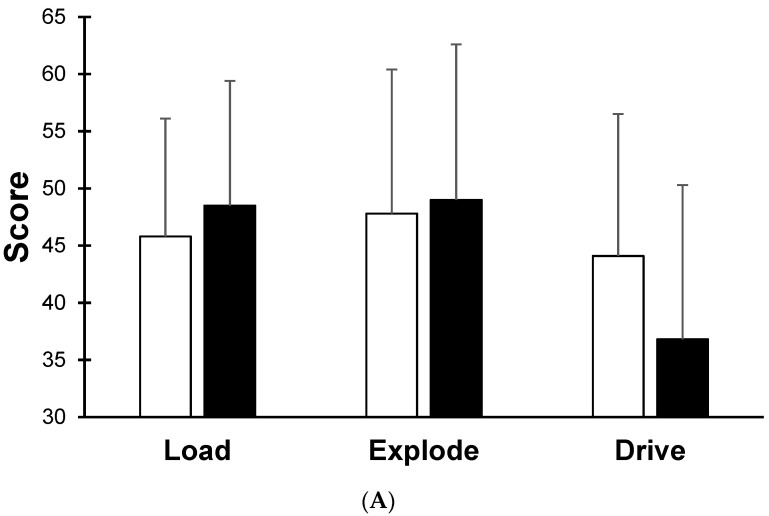
Two-leg SPARTA profiles for CON (**A**) and EXP (**B**) prior to (white bars) and following (black bars) 5-weeks of training.

**Table 1 sports-11-00008-t001:** Participants’ baseline characteristics.

	CON (*n* = 9)	EXP (*n* = 8)
**Age** (years)	30.9 ± 9.4	24.9 ± 3.8
**Height** (cm)	172.7 ± 6.7	178.9 ± 9.3
**Mass** (kg)	78.4 ± 12.9	75.9 ± 9.5
**BMI** (kg/m^2^)	26.3 ± 4.0	23.8 ± 2.6

Values are mean ± standard deviation; CON = control group; *n* = number of persons; EXP = experimental group; cm = centimetre; kg = kilogram; m = metre.

**Table 2 sports-11-00008-t002:** Results for physical and occupational assessments prior to and following 5-weeks of training.

	CON (*n* = 5–9)			EXP (*n* = 7–8)		
	Week 0	Week 6	Change (%)	Week 0	Week 6	Change (%)
**Muscular endurance**						
Push-ups in 60-s	29.3 ± 8.5	31.1 ± 8.5	7.5 ± 10.8	37.4 ± 10.1	38.1 ± 13.9	0.8 ± 11.8
**Cardiorespiratory fitness**						
Predicted rate of oxygen consumption (mL/kg/min)	41.5 ± 7.6	40.9 ± 6.2	−0.5 ± 6.7	43.8 ± 3.5	43.9 ± 3.7	0.3 ± 4.6
**Muscular strength**						
Squat—3RM (kg)	70.7 ± 13.0	83.9 ± 11.7	20.1 ± 14.2	82.9 ± 25.8	97.1 ± 29.1	17.9 ± 9.0
Strict press—3RM (kg)	39.3 ± 5.3	42.6 ± 3.8	9.1 ± 9.2	39.3 ± 9.3	43.9 ± 12.3	10.8 ± 9.4
Deadlift—3RM (kg)	98.8 ± 19.4	106.3 ± 17.7	8.4 ± 6.8	92.9 ± 26.7	111.4 ± 32.8	20.6 ± 11.9 †
Floor press—3RM (kg)	63.1 ± 13.6	70.0 ± 12.2	12.4 ± 11.4	62.9 ± 21.0	67.9 ± 25.1	6.8 ± 7.8
**Occupational capacity**						
5 km weight loaded march time (min)	50.5 ± 1.4	47.9 ± 2.7	−5.1 ± 3.0	44.1 ± 3.1 *	43.8 ± 3.6	−0.7 ± 4.0 *

Values are mean ± standard deviation; mL = millilitre; kg = kilogram; min = minutes; CON = control group; *n* = number of persons; EXP = experimental group; † *p* = 0.054 vs. CON; * *p* < 0.04 vs. CON.

**Table 3 sports-11-00008-t003:** Results for Sparta Science assessments prior to and following 5-weeks of training.

	CON (*n* = 7–9)			EXP (*n* = 7–8)		
	Week 0	Week 6	Change	Week 0	Week 6	Change
**Two-leg CMJ**						
Sparta total score	78.8 ± 4.9	76.7 ± 5.7	−2.1 ± 4.3	78.1 ± 3.1	79.3 ± 2.0	1.1 ± 2.0
Load score	45.8 ± 10.3	48.5 ± 10.9	2.8 ± 4.0	40.8 ± 5.3	42.7 ± 2.4	1.8 ± 3.9
Explode score	47.8 ± 12.6	49.0 ± 13.6	1.2 ± 5.6	39.5 ± 4.9	43.7 ± 6.5	4.2 ± 5.9
Drive score	44.1 ± 12.4	36.8 ± 13.5	−7.3 ± 9.2	51.2 ± 11.0	44.1 ± 9.6	−7.1 ± 8.2
Jump height (cm)	37.3 ± 10.0	33.6 ± 8.8	−8.9 ± 9.7	35.0 ± 8.2	34.4 ± 7.7	−1.4 ± 9.4
Injury risk score	1.78 ± 1.56	1.89 ± 1.05	0.1 ± 1.5	1.50 ± 0.93	1.25 ± 0.71	−0.3 ± 0.7
**One-leg dominant**						
Sparta total score	66.1 ± 2.5	66.4 ± 3.0	0.3 ± 1.1	68.0 ± 1.8	67.1 ± 4.6	−0.6 ± 5.5
Load score	41.0 ± 13.4	39.5 ± 11.6	−1.5 ± 3.5	34.6 ± 4.2	36.7 ± 6.0	1.9 ± 7.5
Explode score	28.3 ± 11.3	29.9 ± 9.9	1.7 ± 4.7	21.9 ± 2.8	21.4 ± 6.7	0.0 ± 7.5
Drive score	32.7 ± 15.4	27.4 ± 13.3	−5.3 ± 9.4	47.5 ± 5.4	57.0 ± 32.9	7.9 ± 37.4
Jump height (cm)	17.3 ± 3.5	17.3 ± 3.9	0.3 ± 10.7	17.9 ± 4.0	18.7 ± 4.6	4.1 ± 9.4
Injury risk score	2.14 ± 1.57	1.86 ± 1.57	−0.3 ± 0.8	2.71 ± 1.38	3.57 ± 1.51	0.3 ± 2.4
**One-leg non-dominant**						
Sparta total score	65.8 ± 1.8	65.1 ± 6.3	−0.6 ± 6.5	67.9 ± 3.2	69.1 ± 3.1	1.3 ± 3.2
Load score	36.6 ± 8.4	38.3 ± 8.9	1.7 ± 4.1	35.7 ± 3.1	36.5 ± 3.2	0.8 ± 3.4
Explode score	23.6 ± 9.0	26.6 ± 9.1	2.9 ± 5.4	21.8 ± 4.6	24.1 ± 5.2	2.3 ± 5.0
Drive score	47.7 ± 27.0	28.6 ± 16.5	−19.1 ± 21.1	49.3 ± 12.8	43.4 ± 12.8	−5.9 ± 14.5
Jump height (cm)	18.1 ± 5.8	15.5 ± 4.1	−11.0 ± 18.0	17.4 ± 4.4	17.6 ± 4.0	1.8 ± 9.3
Injury risk score	3.00 ± 1.51	2.25 ± 1.83	−0.8 ± 2.4	2.50 ± 1.77	2.50 ± 1.77	0.0 ± 2.1
**Balance**						
Dominant (D) leg score	52.6 ± 5.3	45.2 ± 8.5	−7.4 ± 11.4	55.3 ± 8.2	49.3 ± 7.8	−5.9 ± 3.6
Non-dominant (ND) leg score	51.6 ± 6.5	48.3 ± 7.5	−3.2 ± 10.4	54.7 ± 7.9	51.1 ± 7.5	−3.6 ± 3.6
Asymmetry ratio (D:ND)	1.03 ± 0.08	0.95 ± 0.23	−0.07 ± 0.20	1.01 ± 0.07	0.96 ± 0.03	−0.04 ± 0.08
**Plank**						
Dominant (D) arm score	45.5 ± 2.4	43.7 ± 2.7	−1.7 ± 2.7	54.5 ± 18.7	45.0 ± 2.3	−9.6 ± 18.9
Non-dominant (ND) arm score	47.2 ± 3.4	45.9 ± 3.0	−1.3 ± 1.6	61.9 ± 21.7	46.6 ± 2.9	−15.3 ± 19.7
Asymmetry ratio (D:ND)	0.97 ± 0.04	0.96 ± 0.09	−0.01 ± 0.06	0.92 ± 0.21	0.97 ± 0.07	0.05 ± 0.16

Values are mean ± standard deviation for scores, ratios and changes in scores and ratios; jump height is expressed in centimetres (cm) with change as a percentage of Week 0 value; CON = control group; *n* = number of persons; EXP = experimental group.

## Data Availability

The data underlying this article will be shared on reasonable request to the corresponding author.

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
