# Peer review of "Impact of a 5-Week Individualised Training Program on Physical Performance and Measures Associated with Musculoskeletal Injury Risk in Army Personnel: A Pilot Study"

_sports, 2023, doi:10.3390/sports11010008_

Round 1

Reviewer 1 Report

General comments

The authors have clearly stated that the purpose of the study was to examine the feasibility and effect of an individualised training program on physical performance and musculoskeletal injury risk factors in army personnel. The paper is well-written, easy to follow and adds merit to the vital role of physical activity and exercise in these populations. Given this approach, this work can enhance future attempts in similar research area. However, I have highlighted a few suggestions and concerns in my specific comments section (below) that need to be addressed before considering whether this work should be published or not.

Specific comments

INTRODUCTION

- I suggest adding a sentence about the beneficial role of physical activity in general health of adults according to the latest guidelines by the World Health Organization (1).

- A statement about the popularity of bodyweight training, functional fitness training, and strength training with free weights widely used in physical training programs for military personnel worldwide (including Australia) according to the latest reports published by the American College of Sports Medicine (2), it could be a useful addition.

Suggested References:

1.      Bull FC, Al-Ansari SS, Biddle S, Borodulin K, Bumanat MP, Cardonal G, et al. World Health Organization 2020 guidelines on physical activity and sedentary behaviour. Br J Sports Med 2020; 54(24): 1451-1462.

2.      Kercher VM, Kercher K, Bennion T, Levy P, Alexander C, Amaral PC, et al. 2022 Fitness Trends from Around the Globe. ACSMs Health Fit J 2022; 26(1): 21-37.

RESULTS

- Dropout rates should be mentioned.

- Significance with p values should presented in Table 2 and Figure 1 (if applicable).

DISCUSSION

- Strengths and limitations should be presented in a separate paragraph.

- In conclusions, you should underline the main findings and suggest future research attempts in this area while highlighting potential practical implications.

Reviewer 2 Report

First of all, thank you for the opportunity to review this article.

Below are some recommendations for authors:

Introduction

-Punctuation is required at the end of the first paragraph.

-There is double spacing between most of the paragraphs of the article.

-In the second paragraph there is a punctuation before the bibliographic citations.

-"Whilst" instead of "While"

-"purpoted"?? Third paragraph

-I don't think the formulation of the objective and the last sentence of the introduction is well thought out. The objective is to understand the contribution that force platforms can make to developing an individualised training programme to reduce the risk of injury, however, the last sentence points out the relevance of individualised programmes for this population so I think the study should focus on the development and implementation of the training programme. Please rewrite these last lines to clarify and facilitate the readers' understanding.

Methods

-I would recommend the authors to include new sub-sections under "Methods". Because it makes it very difficult to read and very hard to follow.

-Measures of means and standard deviations should be presented either in a table or in a table between brackets.

-I would recommend adding citations in the tests used for assessment and monitoring, it must be demonstrated that the battery has been standardised beforehand.

-I don't understand how 2 CMJs are performed and later it is stated that there are 4 CMJs with 15 seconds rest. Must be clarified.

-A large number of assessment tests are used, but none have been described accurately.

-The citation after 3RM protocol is in a different format.

-The specifications of the Spartan software should be stated the first time it is mentioned.

-The section on statistical analysis is very brief and should be applied.

Results

-The significance symbol for deadweight is not correct.

-What is "Drive" score? For this reason, the measures section needs to be developed further.

- Figure 1 does not provide relevant information, it should be deleted.

Discussion

-Of course, force platforms are a reliable and valid method of evaluation, but it is not even specified how the height of the jumps is calculated (time or speed of exit) or with which programme the data is processed.

-the term "Drive" is defined in the discussion, it should be carried out in Methods.

-The term "explode" is also defined in this section. It cannot be assumed that these concepts are familiar to the reader.

-In the results, no changes were observed after the training programme in the force platform assessments. Therefore, it cannot be asserted in the discussion that the experimental group has improved.

Conclusions

-The conclusions are well written and cautious, very correct in this section.

Reviewer 3 Report

Dear Authors

You have written an interesting paper. However, some parts need to be addressed for greater clarity.

Please put dots after the references .[4, 6, 7]

In the introduction, some more description of the asymmetries measured by force plates in military personnel is needed - what was found. I suggest some additional literature/current studies where force plates and balance were used :

doi: 10.1519/JSC.0000000000003344; http://dx.doi.org/10.1136/bmjmilitary-2021-001899

This could strengthen your rationale.

Methods:

Please add subsections in the methods for greater clarity.

Why did you use the CMJ with arm swing? This was not touched on in the introduction. A sentence or 2 would be good to show the usage in military personnel.

CMJ with only 15s rest? Usually, it is at least 1 min to allow for adequate recovery. Back up this and/or add this in the limitation section. Did they have any demonstrations and/or trial attempts? Report

How did you determine the dominant leg? Report

Please provide a figure of all tests performed and their sequence with breaks for greater understanding and visual presentation of tests.

3RM) protocol. 9 / Incorrect referencing

What was the order of training sessions? report

What was the pace of the aerobic sessions or HR intensity? Report

Also it is not clear what is meant by 400-800m. Was the distance in between or was one distance 400m and then 800m. Please be clear and specific

24kg loaded - how was the load administered - report

The rest of the strength training is confusing and hard to follow. Divide it in subsections and present it more clearly.

How was the normality of data checked? report

The limitations of the study could be expanded. The participants were from different branches of the military where we know the physical specificity and occupation load are different.

Overall a promising study, that needs more clearer methods section for increased reproducibility of this study. Therefore I recommend a major revision.

Kind regards

Round 2

Reviewer 2 Report

Methods

- In subsection Countermovement jump assessments there is a typo, one-arm plank.

Author Response

Dear Associate Editor and Reviewers,

The authors would like to thank you for your comments to improve the manuscript. 

We have corrected the typo in subsection Countermovement jump assessments from "one-am plank" to "one-arm plank" (line 213)

With Thanks,

Corresponding author (on behalf of all authors).

Reviewer 3 Report

Dear Authors,

Thank you for addressing all of my comments. In my opinion, the quality of the manuscript improved.

Overall I don't agree with the Sparta procedure of a 15s break. Therefore, please try to report the validity and reliability of this system in the manuscript.

Regarding the photo material. It is for your benefit to improve the clarity of your manuscript. Perhaps you have similar photos that you could use for this manuscript? Overall I would greatly recommend including them, however as we all know how publishing goes and the response from the other journal might not be as fast, I will not insist on it.

Overall, the paper is acceptable for publication after a minor revision.

Kind regards

Author Response

Dear Associate Editor and Reviewers,

The authors would like to thank you for your comments to improve the manuscript. A description of the authors’ responses to each comment is provided below, including the location of the changes in the manuscript.

With Thanks,

Corresponding author (on behalf of all authors).

----------------------------------------------------------------------------------------

Overall I don't agree with the Sparta procedure of a 15s break. Therefore, please try to report the validity and reliability of this system in the manuscript.

RESPONSE: We have included a section in the introduction which specifies the reliability of the Sparta system in a military personnel using the CMJ assessment with a 15s break between jumps (lines 113-116).

“Based on the CMJ outcomes, which have demonstrated excellent test-retest reliability in military personnel (ICC>0.90) [18], and implied imbalances, an individualised strength program is then recommended to improve these imbalances and decrease the risk of future MSKI [19, 23, 24].”

Regarding the photo material. It is for your benefit to improve the clarity of your manuscript. Perhaps you have similar photos that you could use for this manuscript? Overall I would greatly recommend including them, however as we all know how publishing goes and the response from the other journal might not be as fast, I will not insist on it.

RESPONSE: Whilst we understand the benefit of including photo material to improve the clarity of the manuscript, we unfortunately we do not have similar photos that could be used. The authors wish to thank you for your understanding regarding this matter.